# Increased Mild Vaccine-Related Side Effects and Higher Specific Antibody Titers in Health Care Workers with Previous SARS-CoV-2 Infection after the mRNA BNT162b2 Vaccine

**DOI:** 10.3390/vaccines10081238

**Published:** 2022-08-02

**Authors:** Ludovica Ferrari, Mirko Compagno, Laura Campogiani, Elisabetta Teti, Tiziana Mulas, Davide Checchi, Grazia Alessio, Federica Caldara, Luigi Coppola, Giuseppe De Simone, Laura Ceccarelli, Ilaria Spalliera, Pietro Vitale, Sandro Grelli, Massimo Andreoni, Loredana Sarmati, Marco Iannetta

**Affiliations:** 1Infectious Diseases Clinic, Policlinic of Tor Vergata, 00133 Rome, Italy; ludovica.ferrari@ptvonline.it (L.F.); mirko.compagno@ptvonline.it (M.C.); laura.campogiani@ptvonline.it (L.C.); elisabetta.teti@ptvonline.it (E.T.); tiziana.mulas@ptvonline.it (T.M.); davide.checchi@ptvonline.it (D.C.); grazia.alessio@ptvonline.it (G.A.); federica.caldara@ptvonline.it (F.C.); luigi.coppola@ptvonline.it (L.C.); giuseppe.desimone@ptvonline.it (G.D.S.); laura.ceccarelli@ptvonline.it (L.C.); ilaria.spalliera@ptvonline.it (I.S.); pietro.vitale@ptvonline.it (P.V.); 2Department of Experimental Medicine, University of Rome Tor Vergata, 00133 Rome, Italy; grelli@med.uniroma2.it; 3Virology Unit, Policlinic of Tor Vergata, 00133 Rome, Italy; 4Department of Systems Medicine, University of Rome Tor Vergata, Via Montpellier 1, 00133 Rome, Italy; andreoni@uniroma2.it (M.A.); marco.iannetta@uniroma2.it (M.I.)

**Keywords:** vaccines, SARS-CoV-2, COVID-19, healthcare workers, BNT162b2, side effects, antibody titer

## Abstract

Background: to evaluate whether prior SARS-CoV-2 infection affects side effects and specific antibody production after vaccination with BNT162b2. Methods: We included 1106 health care workers vaccinated with BNT162b2. We assessed whether prior SARS-CoV-2 infection affects the number and type of side effects and performed a nested case–control analysis comparing plasma levels of specific IgG titers between SARS-CoV-2-naïve and previously infected subjects after the first and the second vaccine doses. Results: After the first dose, SARS-CoV-2-naïve subjects experienced side effects more often than SARS-CoV-2 naïve subjects. Individuals with prior SARS-CoV-2 infection more often reported pain at the injection site, weakness, and fever than SARS-CoV-2-naïve subjects. After the second dose, the frequency of side effects was similar in the two groups. All subjects with prior SARS-CoV-2 infection developed either a high (>100 AU/mL) or intermediate (10–100 AU/mL) antibody titer. Among SARS-CoV-2-naïve subjects, the majority developed an intermediate titer. After the second dose, a high (>2000 AU/mL) antibody titer was more common among subjects with prior SARS-CoV-2 infection. Conclusions: vaccine-related side effects and a higher anti-SARS-CoV-2-RBD IgG titer were more common in subjects with previous infection than in SARS-CoV-2-naïve after the first, but not after the second dose of the BNT162b2 vaccine.

## 1. Introduction

Severe acute respiratory syndrome coronavirus 2 (SARS-CoV-2) is the causative agent of coronavirus disease-19 (COVID-19), which has led to an estimated 4.8 million deaths worldwide as of December 2021. Since its genetic sequence became available in January 2020, the development and testing of vaccines directed against SARS-CoV-2 has grown at an unprecedented pace. The distribution of vaccines, together with nonpharmacological interventions, has proven the only possible strategy to control the pandemic; nonetheless, the achievement of global vaccine coverage is hindered by both socioeconomic and biological hurdles. The safety and efficacy of several vaccines were tested in phase III randomized controlled trials (RCTs), resulting in their emergency approval by the United States Food and Drug Administration (FDA) and the European Medicines Agency (EMA). Messenger RNA (mRNA) vaccines became available in December 2020, and the BNT162b2 Pfizer BioNTech was the first vaccine granted an Emergency Use Authorization by the FDA [1], and the EMA [2].

In the RCTs, local and systemic acute reactions were frequently reported, especially after the second vaccine dose, and were usually mild and transient. In BNT162b2 RCTs, the most common local side effect was pain at the site of injection, and the most common systemic side effects were fatigue, headache, and new onset or worsening of muscle pain [3].

One open question is whether a prior SARS-CoV-2 infection affects side effects and immunogenicity of the BNT162b2 vaccine. In this study, we compared solicited side effects and specific antibody production after the first and second dose of the BNT162b2 vaccine in subjects with a previous RT-PCR-confirmed SARS-CoV-2 infection and SARS-CoV-2-naïve subjects in a cohort of health care workers (HCWs).

## 2. Materials and Methods

### 2.1. Study Design and Participants

This was a retrospective observational cohort study with a case–control study nested to the cohort, involving health care workers (HCWs) who received vaccination against COVID-19 at Tor Vergata University Hospital (Rome, Italy). It consisted of two phases: a retrospective collection of vaccine side effects reported by HCWs and hospital staff members who received COVID-19 vaccination with the BNT162b2 (Comirnaty^®^) vaccine, and a case–control study, comparing vaccine side effects in individuals with and without prior SARS-CoV-2 infection. In both phases, a subgroup of subjects gave consent for blood sampling; hence, plasma levels of class G immunoglobulin (IgG) directed toward the SARS-CoV-2 receptor-binding domain (RBD) were measured. The study was approved by the local Ethics Committee (Independent Ethics Committee Fondazione PTV, Policlinico Tor Vergata, protocol register number 40/21). Written informed consent was obtained from the participating subjects. The study was conducted in accordance with the principles of the Declaration of Helsinki.

In the first phase, HCWs and hospital staff members who received the Comirnaty^®^ vaccine from 31 December 2020 to 28 January 2021 according to the approved schedule were included. Data on previous SARS-CoV-2 infection, demographics, smoking habits, and self-reported comorbidities were collected in a structured questionnaire administered at the time of the second dose injection (21 days after the first dose). Simultaneously, side effects that developed after the first vaccine dose were collected through the same questionnaire. Subjects for whom data on the SARS-CoV-2 prior infection were not available were later eliminated from the study population. Analyses were conducted in the overall included population and after stratification according to prior SARS-CoV-2 infection status. In a subgroup of patients who consented to blood sampling, we assessed plasma levels of class G immunoglobulins (IgG) directed toward the SARS-CoV-2 receptor-binding domain (RBD).

The second phase of the study consisted of a case–control study nested to the cohort to analyze the frequency and type of side effects in subjects without a prior SARS-CoV-2 infection (controls) compared to those with a history of SARS-CoV-2 infection (cases). All subjects with a previous RT-PCR-confirmed SARS-CoV-2 infection were included as cases and were matched 1:2 with controls based on age, sex, smoking habits (previous or current smoker vs. never smoker) and comorbidities. Three weeks after the second dose, subjects included in the analysis were contacted by telephone and a structured questionnaire on side effects was administered. Again, a subgroup gave consent to give a blood sample to assess their anti-SARS-CoV-2 S-RBD IgG titer. Blood sampling was performed within 60 days after the second vaccine dose.

Prior SARS-CoV-2 infection was defined by at least one nasopharyngeal swab positive for SARS-CoV-2 RNA detection. In Figure 1, an algorithm of the study populations’ inclusion and exclusion criteria is presented.

### 2.2. Laboratory Test

Nasopharyngeal swabs were processed according to standard diagnostic protocols using the GeneFinderTM COVID-19 Plus RealAmp Kit, ELITech AllplexTM 2019-nCoV Assay (Seegene). IgG/IgM antibodies toward the SARS-CoV-2 spike RBD region were detected with a commercial Elecsys^®^ (Roche) immunoassay according to the manufacturer’s instructions. Anti-SARS-CoV-2 RBD IgG titers assessed after the first vaccine doses were classified as low (0–10 AU/mL), intermediate (10–100 AU/mL) and high (>100 AU/mL). Blood samples collected after the second vaccine dose were appropriately diluted to enable anti-RBD IgG quantification of up to 2000 AU/mL. Therefore, anti-SARS-CoV-2 RBD IgG titers after the second vaccine dose were classified as low (0–100 AU/mL), intermediate (100–2000 AU/mL) and high (>2000 AU/mL).

### 2.3. Statistical Analysis

Statistical analyses were performed with JASP version 0.14.1 (University of Amsterdam, Amsterdam, The Netherlands). A two-tailed Mann-Whitney U-test and a Chi-squared test were used for between-group comparisons of continuous values and proportions, respectively. Statistical significance was defined as *p* < 0.05.

## 3. Results

### 3.1. Overall Population

A total of 1132 HCWs received the first dose of the BNT162b2 vaccine at our institution. Of them, 26 were excluded from the study because of missing information on previous SARS-CoV-2 infection, with a resulting study population of 1106 individuals (Figure 1). The baseline characteristics of the study population are presented in Table 1. The median age was 42 years (interquartile range 31–51), 441 (39.8%) were males, and 327 (29.6%) were active smokers; hypertension was the most common comorbidity (10.2%). Eighty-three (7.5%) subjects had a history of SARS-CoV-2 infection.

Comparing subjects with and without a prior SARS-CoV-2 infection, active smokers were more common among SARS-CoV-2-naïve subjects than among subjects with a previous SARS-CoV-2 infection (30.6% vs. 16.9%, respectively, *p* = 0.008); no significant difference was recorded in the prevalence of comorbidities (*p* > 0.05).

### 3.2. Number and Type of Solicited Side Effects in the Overall Population

Overall, 752 (68.0%) subjects reported at least one side effect after the first vaccine dose, all of which occurred within the first week from the vaccine administration. The number and type of solicited vaccination-related side effects after the first dose are presented in Appendix A, respectively. The BNT162b2 vaccine was well tolerated and no severe side effects were reported.

After the first dose, subjects more often reported one (501/1106 individuals, 45.3%) or no side effects (354/1106, 32.0%) (Appendix A). The most common solicited side effects were pain at the injection site (704/1106, 63.6%), muscle or joint pain (133/1106, 12.0%), weakness (114/1106, 10.3%), and fever (34/1106, 3.1%) (Table 2).

To address the influence of prior SARS-CoV-2 infection on vaccination-related side effects, we compared the number and type of solicited side effects in individuals with a RT-PCR-confirmed prior SARS-CoV-2 infection and SARS-CoV-2-naïve subjects. After the first dose, subjects with a previous SARS-CoV-2 infection reported to have experienced side effects more frequently than SARS-CoV-2 naïve subjects (18/83, 21.7% vs. 336/1023, 32.8%, *p* = 0.038). In addition, individuals with a history of SARS-CoV-2 infection more frequently reported the co-occurrence of two (16/83, 19.3%) or three or more (13/83, 15.7%) side effects than SARS-CoV-2 naïve subjects, who had more frequently one (465/1023, 45.5%) or no (336/1023, 32.8%) side effects (*p* = 0.021). Regarding side effect type, subjects with a prior SARS-CoV-2 infection more often reported pain at the injection site (75.9% vs. 62.7%, *p* = 0.016), weakness (22.9% vs. 9.3% *p* < 0.001) (Appendix A), and fever (9.6% vs. 2.5% *p* < 0.001) than subjects without a prior SARS-CoV-2 infection (Table 2).

### 3.3. Case–Control Analysis: Number and Type of Solicited Side Effects

To account for the effects of demographic and comorbidities as potential confounders, we analyzed the number and type of side effects after matching subjects with a history of SARS-CoV-2 infection (cases) in a 1:2 ratio with SARS-CoV-2-naïve subjects (controls) based on age, sex, smoking habits, and comorbidities, including a total of 249 subjects (83 cases and 166 controls).

The number and type of solicited vaccination-related side effects, after the first and second doses, after case–control matching are presented in Appendix A and Table 3, respectively.

After the first dose, the proportion of subjects reporting at least one side effect did not differ between cases and controls (18/83, 21.7% vs. 46/166, 27.7%, *p* = 0.36). However, cases more frequently reported the co-occurrence of two (16/83, 19.3%), three, or more (13/83, 15.7%) side effects than control subjects, who had more often one (87/166, 52.4%) or no (46/166, 27.7%) side effects (*p* = 0.028) (Appendix A and Figure 2). Weakness and fever occurred more frequently among cases than in controls [19/83 (22.9%) vs. 11/166 (6.6%), *p* < 0.001; 8/83 (9.6%) vs. 5/166 (3.0%), *p* < 0.027, respectively]. In addition, headache tended to be more common among cases than controls, although the difference was not statistically significant [10/83 (12.0%) vs. 10/166 (6.0%), *p* = 0.099] (Table 3).

After the second dose, 189 subjects (79%) reported at least one side effect. The incidence of side effects (20/82, 24.4% vs. 30/157 19.1%, *p* = 0.40) as well as their number (Appendix A and Figure 3) and type (Table 3) did not differ between the two groups, with the most common systemic reaction being muscle or joint pain in both groups (29.3% cases vs. 38.2% controls, *p* = 0.169). Pain in the injection site was significantly more frequent in controls [36/82, 43.9% cases vs. 90/157 57.3% controls, *p* = 0.048].

### 3.4. Anti-SARS-CoV-2 RBD IgG Titer

We sought to assess the impact of a history of SARS-CoV-2 infection on the antibody response to the mRNA BNT162b2 vaccine.

We measured the anti-SARS-CoV-2 RBD IgG titer in 224 individuals, 3 weeks after the first vaccine dose (Table 4.). Anti-SARS-CoV-2 RBD IgG titers after the first vaccine dose were classified as low (0–10 AU/mL), intermediate (10–100 AU/mL), or high (>100 AU/mL). All subjects with a previous SARS-CoV-2 infection developed either a high (50/62, 80.6%) or intermediate (12/62, 19.4%) antibody titer. Among SARS-CoV-2-naïve subjects, the majority (120/162, 74.1%) developed an intermediate antibody titer. None of the subjects with a history of SARS-CoV-2 infection developed a low antibody titer, as opposed to the 25/162 (15.4%) subjects without a prior SARS-CoV-2 infection (*p* < 0.001).

After a median of 19 days (IQR 15-29) following administration of the second vaccine dose, the antibody titer was quantified in 197 individuals (Table 4), and anti-SARS-CoV-2 RBD IgG titers were classified as low (0–100 AU/mL), intermediate (100–2000 AU/mL), and high (>2000 AU/mL). Blood sampling was performed within 60 days after the second vaccine dose. The median interval between the second vaccine dose and antibody quantification was 23 days (range 14–54).

The majority of cases (38/46, 82.6%) and controls (129/151, 85.4%) developed an intermediate antibody titer. A high antibody titer was more frequent among subjects with a prior SARS-CoV-2 infection than among the controls (6/46, 13.0% vs. 6/151, 4.0%, respectively, *p* = 0.043). Conversely, a low antibody titer was less common among the cases than among the controls (2/46, 4.3% vs. 16/151, 10.6%, *p* = 0.043).

## 4. Discussion

The results of this study indicate that subjects with a previous SARS-CoV-2 infection developd more vaccine-related side effects and a higher anti-SARS-CoV-2 RBD IgG titer compared with SARS-CoV-2-naïve individuals after the first, but not the second dose of the BNT162b2 vaccine. We focused on the BNT162b2 vaccine since this was the only approved COVID-19 vaccine when the study was conducted and the HCWs vaccination campaign in Italy was performed primarily with this vaccine.

HCWs are considered at high risk of SARS-CoV-2 infection and were therefore prioritized for vaccination in the first months of vaccine distribution. The majority of HCWs joined the vaccination campaign against SARS-CoV-2, which in Italy started on 27 December 2020 [4]. HCWs have a low burden of comorbidities and a relatively high prevalence of a prior SARS-CoV-2 infection and thus represent an ideal population in which to assess the side effects and immunogenicity of the BNT162b2 vaccine [5,6,7,8,9,10].

The pivotal phase III trial of BNT162b2 demonstrated that the vaccine is safe and well tolerated [3]. In our cohort, no severe adverse effects were reported, confirming the overall tolerability of the vaccine. In the trial, the proportion of vaccinated subjects reporting local reactions such as pain, redness, and swelling at the injection site was similar after the two doses, whereas systemic events such as fever were more common and more severe after the second dose than after the first dose [9]. One open question is whether and to what extent a prior SARS-CoV-2 infection affects the frequency and severity of side effects of the BNT162b2 vaccine. In our study, we observed that after the first dose, prior SARS-CoV-2 infection was associated with a higher risk of developing both local and systemic reactions. Specifically, the most common side effect was pain at the site of injection, which was reported by 75.9% and 68.1% of patients with and without a prior SARS-CoV-2 infection, respectively (*p* = 0.201). Similarly, weakness and fever were more frequent among patients with a prior SARS-CoV-2 infection (*p* < 0.001). Other side effects, such as headache and diarrhea, were rare and did not differ between the two groups. Patients with a history of SARS-CoV-2 infection reported on average a greater number of side effects than SARS-CoV-2-naïve subjects after the first vaccine dose, whereas after the second dose, the number and type of side effects did not differ, regardless of the prior SARS-CoV-2 infection status. Case–control matching of subjects with and without a prior SARS-CoV-2 infection allowed us to eliminate confounding factors that might interplay in the immune response, such as age, sex, and comorbidities, when analyzing the response to the second vaccine dose. After case–control matching, the number of solicited side effects was greater in cases than in controls after the first vaccine dose but was comparable after the second dose between the two groups. These observations likely reflect a more intense immune response in individuals whose immune system had already been primed during a SARS-CoV-2 infection.

Accordingly, individuals with a history of SARS-CoV-2 infection developed a higher IgG (S-RBD) antibody titer than SARS-CoV-2-naïve subjects after the first dose (*p* < 0.001). The vast majority (80.6%) of subjects with a previous infection developed a high (>100 AU/mL) antibody titer after the first dose. After the second dose, the antibody response was similar in previously infected and uninfected individuals, even if previously infected subjects maintained a greater prevalence of high antibody titer response after vaccination. These results are in line with one previous report that considered 51 HCWs in London and indicated that among subjects with a previous SARS-CoV-2 infection, vaccination increased anti-S titers more than 140-fold from pre-vaccine levels, which is markedly higher compared to the SARS-CoV-2-naïve individuals [5]. Indeed, one vaccine dose was sufficient to boost both cellular and humoral responses in subjects who had already recovered from COVID-19, whereas the second vaccine dose proved necessary in naïve subjects to enhance the immune response [11]. This reflects the effective reactivation of the humoral and cellular immunological memory directed against the SARS-CoV-2 Spike protein in subjects that were previously exposed to the virus [11], which translates into a significant reduction in the risk of recurrent SARS-CoV-2 infection [12]. Additionally, in line with our results, it has been reported that the antibody response after two doses of the BNT162b2 vaccine is similar to the response after a single dose in individuals with a prior SARS-CoV-2 infection and, also lending support to our results, antibody titer in subjects with a previous SARS-CoV-2 infection exceeded the median antibody titers measured in participants without pre-existing immunity after the second vaccine dose by more than six-fold [13], and side effects are more evident for those with a previous infection after the first dose [14], but not after the second dose [10].

After SARS-CoV-2 infection, an antibody response directed against the virus can be detected in most infected individuals. The magnitude of the antibody response, but not its time course, correlates with the severity of the disease [15], and levels of neutralizing antibodies decline to low levels after a low-severity disease. In our study, both SARS-CoV-2-naïve and previously infected individuals developed a high antibody titer after the second vaccine dose, in line with previous studies demonstrating that two doses of the BNT162b2 vaccine induce high levels of SARS-CoV-2 IgG [8]. Although a recent study on 3500 HCWs showed that a longer interval between infection and the first vaccine dose may enhance the antibody response [16], it remains to be fully elucidated whether the kinetics and longevity of the antibody response are affected by a previous SARS-CoV-2 infection.

Our study has some limitations. We defined prior SARS-CoV-2 infection as the positivity of at least one nasopharyngeal swab for the detection of SARS-CoV-2 RNA. However, during the early stages of the pandemic, molecular testing was offered mostly to symptomatic individuals, which might have led to an underestimation of SARS-CoV-2 infection in asymptomatic subjects. Nonetheless, the majority of HCWs were periodically screened regardless of the presence of symptoms. Our study cohorts included individuals between 21 and 69 years old, with a median age of 42 years. In clinical trials, patients above 65 years of age developed a lower antibody titer than younger individuals; therefore, our results may not apply to elderly individuals. We measured anti-SARS-CoV-2 RBD IgG titers, however, we did not quantify antibody levels above 100 AU/mL and 2000 AU/mL after the first and second doses, respectively, due to technical limitations of the assay employed. Furthermore, the study of serum neutralizing activity, which represents the most reliable correlate of protection from a SARS-CoV-2 infection conferred by COVID-19 vaccines, is lacking. Finally, we considered as vaccine-related events all those side effects reported within 3 weeks (21 days) from vaccine administration, but we did not categorize side effects as early or late, thus missing detailed information on the time course of these side effects.

## 5. Conclusions

Our results demonstrate that subjects with a history of a SARS-CoV-2 infection develop more vaccine-related side effects and a higher anti-SARS-CoV-2 RBD IgG titer than SARS-CoV-2-naïve individuals after the first dose of the BNT162b2 vaccine. After the second vaccine dose, the side effects are comparable, but a trend toward a higher antibody titer is observed in subjects with a prior SARS-CoV-2 infection.

These observations likely reflect a more intense immune response in individuals whose immune systems had already been primed toward SARS-CoV-2 and are efficiently boosted upon exposure to the SARS-CoV-2 S protein after vaccination. These results suggest that the antibody response after a single dose of the BNT162b2 vaccine in individuals with a prior SARS-CoV-2 infection is comparable to a two-dose vaccine course in SARS-CoV-2 naïve subjects.

## Figures and Tables

**Figure 1 vaccines-10-01238-f001:**
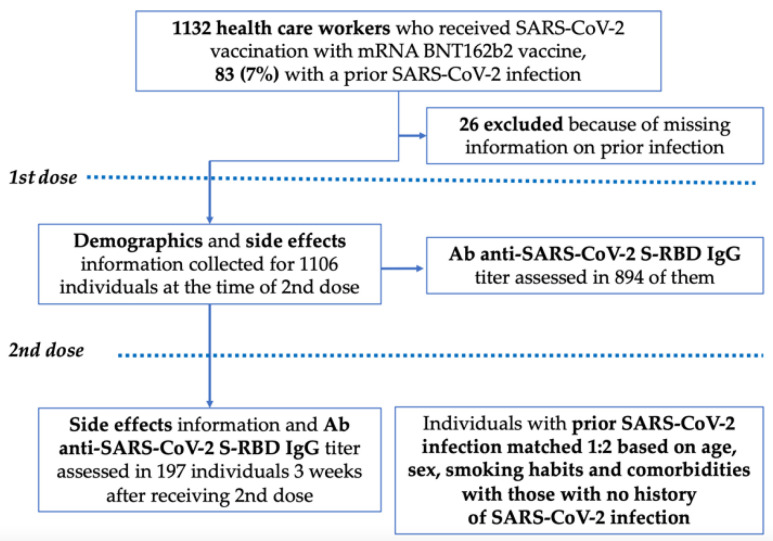
Study design. Flow chart describing the population of health care workers (HCW) analyzed in the study. For the case–control analysis, 249 subjects were enrolled (83 cases and 166 controls).

**Figure 2 vaccines-10-01238-f002:**
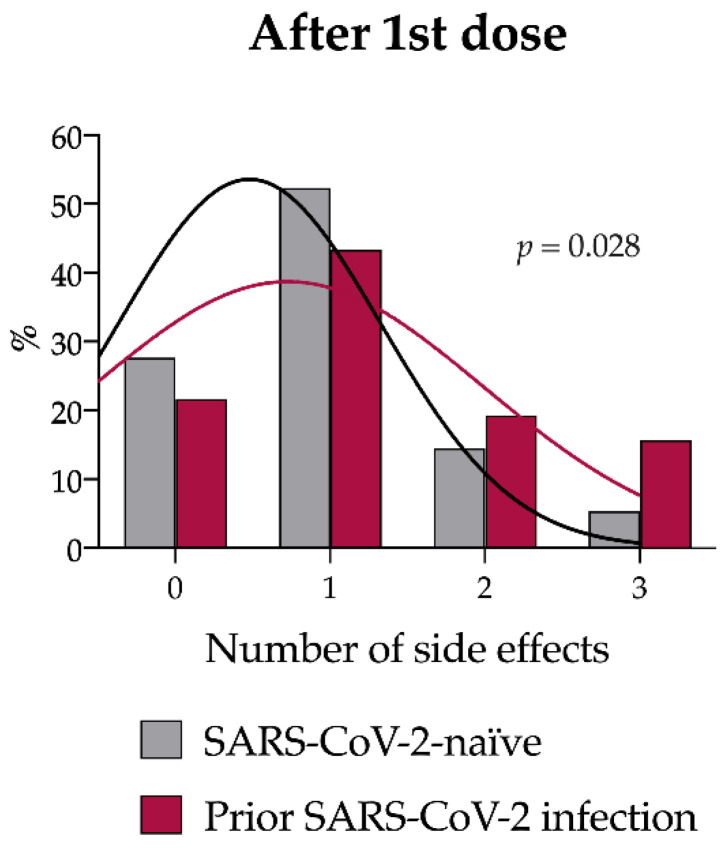
Number of solicited side effects after the first vaccine dose in the study population after case–control matching. Number of side effects reported by SARS-CoV-2 naïve (gray) and previously infected (magenta). A right shift in the distribution of the number of side effects was observed in SARS-CoV-2 naïve subjects (*p* = 0.028 by Chi-squared test).

**Figure 3 vaccines-10-01238-f003:**
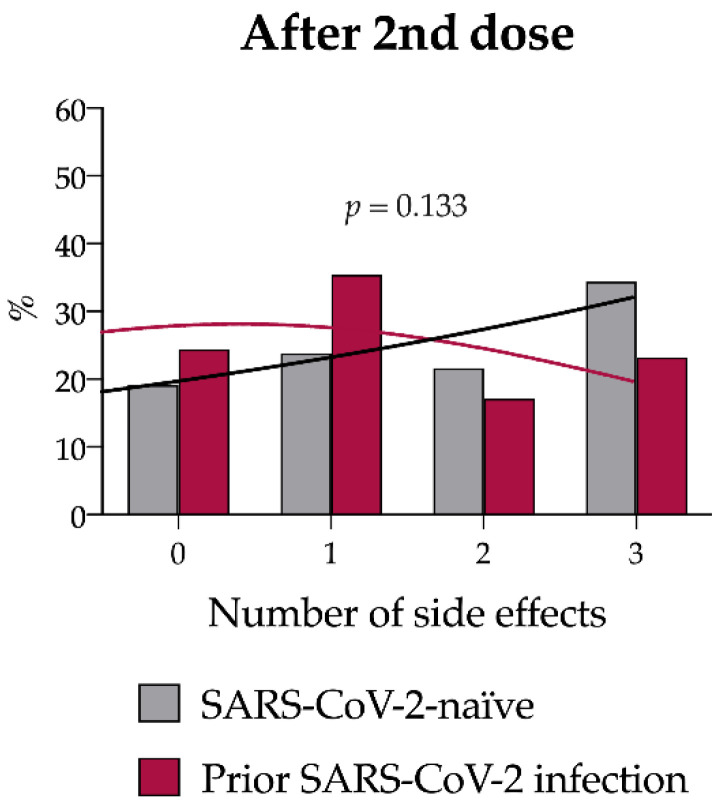
Number of solicited side effects after the second vaccine dose in the study population after case–control matching. Number of side effects reported for SARS-CoV-2 naïve (gray) and previously infected subjects (magenta; *p* = 0.133 by Chi-squared test).

**Table 1 vaccines-10-01238-t001:** Baseline characteristics of the study population, overall and stratified according to prior SARS-CoV-2 infection.

	Overall (n = 1106)	Previous SARS-CoV-2 Infection		*p*
		Yes (n = 83)	No (n = 1023)	
Median age	42 (31–51)	42 (31–51)	40 (30–48)	0.260
Males	441 (39.8%)	36 (43.4%)	405 (39.6%)	0.498
Smokers	327 (29.6%)	14 (16.9%)	313 (30.6%)	**0.008**
Comorbidities				
Hypertension	113 (10.2%)	6 (7.2%)	107 (10.6%)	0.350
Cardiovascular disease	12 (1.1%)	0 (0%)	12 (1.2%)	0.321
Obesity	21 (1.9%)	1 (1.2%)	20 (2.0%)	0.630
Thyroid disorders	110 (9.9%)	6 (7.2%)	104 (10.2%)	0.390
Diabetes	22 (2.0%)	0 (0%)	22 (2.1%)	0.177
Neoplasm	11 (1.0%)	0 (0%)	11 (1.1%)	0.342
Asthma	15 (1.4%)	1 (1.2%)	14 (1.4%)	0.901
Other respiratory diseases	14 (1.3%)	2 (2.4%)	12 (1.2%)	0.332
Autoimmune disorders	30 (2.7%)	1 (1.2%)	29 (2.8%)	0.379
Dyslipidemia	6 (0.5%)	2 (2.4%)	4 (0.4%)	**0.016**
Anemia	4 (0.4%)	1 (1.2%)	3 (0.3%)	0.183
Other	50 (4.5%)	5 (6.0%)	45 (4.4%)	0.493

**Table 2 vaccines-10-01238-t002:** Type of side effects in the study population after the first vaccine dose, overall and stratified according to prior SARS-CoV-2 infection.

	1st Vaccine Dose			
	Overall (n = 1106)	Previous SARS-CoV-2 Infection		*p*
		Yes (n = 83)	No (n = 1023)	
Pain in site of injection	704 (63.6%)	63 (75.9%)	641 (62.7%)	**0.016**
Muscle or joint pain	133 (12.0%)	14 (16.9%)	119 (11.6%)	0.158
Weakness	114 (10.3%)	19 (22.9%)	95 (9.3%)	**<0.001**
Fever	34 (3.1%)	8 (9.6%)	26 (2.5%)	**<0.001**
Headache	102 (9.2%)	10 (12.0%)	92 (9.0%)	0.355
Diarrhea	6 (0.5%)	0 (0%)	6 (0.6%)	0.484
Nausea	8 (0.8%)	0 (0%)	8 (0.8%)	0.419
Dyspnea	4 (0.4%)	1 (1.2%)	3 (0.3%)	0.183
Skin rash	2 (0.2%)	0 (0%)	2 (0.2%)	0.687
Lymphadenopathy	3 (0.3%)	0 (0%)	3 (0.3%)	0.621
Paresthesia	5 (0.5%)	0 (0%)	5 (0.5%)	0.523
Others	17 (1.5%)	2 (2.4%)	15 (1.5%)	0.502

**Table 3 vaccines-10-01238-t003:** Type of side effects after the first and second vaccine doses in the study population after case–control matching, overall and stratified according to prior SARS-CoV-2 infection.

	1st Vaccine Dose				2nd Vaccine Dose			
	Overall (n = 249)	Previous SARS-CoV-2 Infection		*p*	Overall (n = 239)	Previous SARS-CoV-2 Infection		*p*
		Yes (n = 83)	No (n = 166)			Yes (n = 82)	No (n = 157)	
Pain in site of injection	176	63	113	0.201	126	36	90	**0.048**
	(70.7%)	(75.9%)	(68.1%)		(52.7%)	(43.9%)	(57.3%)	
Muscle or joint pain	34	14	20	0.296	84	24	60	0.169
	(13.7%)	(16.9%)	(12.0%)		(35.1%)	(29.3%)	(38.2%)	
Weakness	30	19	11	**<0.001**	87	30	57	0.966
	(12.0%)	(22.9%)	(6.6%)		(34.4%)	(36.6%)	(36.3%)	
Fever	13	8	5	**0.027**	65	18	47	0.188
	(5.2%)	(9.6%)	(3.0%)		(27.2%)	(21.9%)	(29.9%)	
Headache	20	10	10	0.099	42	15	27	0.833
	(8.0%)	(12.0%)	(6.0%)		(17.6%)	(18.3%)	(17.2%)	
Diarrhea	0	0	0	/	2	0	2	0.305
	(0%)	(0%)	(0%)		(0.8%)	(0%)	(1.3%)	
Nausea	1	0	1	0.479	8	2	6	0.573
	(0.4%)	(0%)	(0.6%)		(3.3%)	(2.4%)	(3.8%)	
Dyspnea	1	1	0	0.156	2	1	1	0.639
	(0.4%)	(1.2%)	(0%)		(0.8%)	(1.2%)	(0.6%)	
Skin rash	1	0	1	0.479	1	1	0	0.166
	(0.4%)	(0%)	(0.6%)		(0.4%)	(1.2%)	(0%)	
Lymphadeno-pathy	0	0	0	/	4	1	3	0.692
	(0%)	(0%)	(0%)		(1.7%)	(1.2%)	(1.9%)	
Paresthesia	1	0	1	0.479	3	2	1	0.235
	(0.4%)	(0%)	(0.6%)		(1.2%)	(2.4%)	(0.6%)	
Others	4	2	2	0.476	17	3	14	0.133
	(1.6%)	(2.4%)	(1.2%)		(7.1%)	(3.7%)	(8.9%)	

Information on side effects after second vaccine dose is missing for 10 subjects (1 with prior infection and 9 without).

**Table 4 vaccines-10-01238-t004:** Anti-SARS-CoV-2 RBD IgG titer quantification after the first and second vaccine dose, overall and stratified according to prior SARS-CoV-2 infection.

		1st Vaccine Dose				2nd Vaccine Dose			
Anti-SARS-CoV-2 RBD IgG (AU/mL)	Overall (%) n = 224	Prior SARS-CoV-2 Infection		*p*	Anti-SARS-CoV-2 RBD IgG (AU/mL)	Overall (%) n = 197	Prior SARS-CoV-2 Infection		*p*
		Yes (n = 62)	No (n = 162)				Yes (n = 46)	No (n = 151)	
Low (0–10)	25 (11.2%)	0 (0%)	25 (15.4%)	**<0.001**	Low (0–100)	18 (9.1%)	2 (4.3%)	16 (10.6%)	**0.043**
Inter- mediate (10–100)	132 (58.9%)	12 (19.4%)	120 (74.1%)		Inter- mediate (100–2000)	167 (84.8%)	38 (82.6%)	129 (85.4%)	
High (>100)	67 (29.9%)	50 (80.6%)	17 (10.5%)		High (>2000)	12 (6.1%)	6 13.0%)	6 (4.0%)	

## Data Availability

The data presented in this study are available on request from the corresponding author. The data are not publicly available due to privacy and ethical concerns.

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
