# Peer review of "Increased Mild Vaccine-Related Side Effects and Higher Specific Antibody Titers in Health Care Workers with Previous SARS-CoV-2 Infection after the mRNA BNT162b2 Vaccine"

_vaccines, 2022, doi:10.3390/vaccines10081238_

Round 1
Reviewer 1 Report
In this study, the authors' primary objective is to compare the frequency of events supposedly attributable to the vaccine and the production of IgG antibodies against the BNT162b2 vaccine in a group of health care workers according to the history of SARS-CoV-2 infection.
The authors hypothesize that a history of COVID-19 is associated with a higher frequency of side effects secondary to mRNA BNT162b2 vaccination; as well as the immunogenicity of the vaccine.
1. Title.
a. The title highlights that the events supposedly attributable to the anti-COVID-19 vaccine will be analyzed after the FIRST DOSE of the mRNA BNT162b2 vaccine; as well as the immunogenicity of the vaccine; however, the Objective of the study does not state if it is only after the first dose or also after the second dose, lines 53 to 56; later on, in the content of the Methods and Results it is discovered that not only the evaluations will be made after the first dose, but also after the second dose.
2. Abstract.
a. The knowledge gap stated in the first sentence of the Abstract and the first sentence of line 52 and 53 are inconsistent; it is suggested to make them uniform. The Objective is different from that stated in the Title, see above.
b. In the Results, it is suggested to compare the proportion of vaccine side effects between the group without and with a history of COVID-19. Likewise, it is suggested to compare the mean or median IgG antibody concentration between both groups.
c. In the conclusions it should be clarified that the Title only suggests that side effects will be analyzed only after the first dose of the vaccine and not the second dose.
3. Introduction.
a. The authors are invited to specify whether the events supposedly attributable to the vaccine will only consider those shown after the first dose or also after the second dose; likewise, to specify whether they will only be early events or also late events.
4. Methods.
a. Please state that this is a retrospective cohort study and a case-control study nested to the cohort; as necessary.
b. Please describe the criteria that were used to include subjects susceptible to IgG antibody quantification.
c. First and second paragraphs repeat the study setting, it is suggested to leave only the first paragraph.
d. Why was the adverse event questionnaire administered up to 3 weeks after the first dose of vaccine was administered?
e. Why was it not done immediately to reduce recall bias?
f. Why were the events supposedly attributable to the vaccine not categorized as immediate or delayed?
g. A case-control study is retrospective in nature, there is no point in stating this (... of a retrospective case-control study to ana- 84 lyze the frequency ...); by the way, please describe more fully how the selection of subjects was done in this phase of the study.
h. It seems inconsistent to have quantified IgG antibodies over a time period of up to 60 days; could you describe the mean or median time over which quantification was done?
5. Results.
a. First sentence repeat the date of the study, see above.
b. Why is age presented as median and not mean?
c. Please state the overall incidence of adverse reactions from the vaccine.
d. Table 2 can be deleted and only present in the manuscript the total frequency (%) per group, with and without history of COVID-19 and its respective hypothesis testing.
e. Since the Title states that only adverse events after the first dose of the vaccine are of interest to the authors, it is not necessary to present those of the second dose.
f. Table 4 can be deleted and only present in the manuscript the total frequency (%) per group, with and without history of COVID-19 and its respective hypothesis testing, according to the group that only received the first dose of the vaccine.
g. Table 5, only showing the results of side effects after the first dose of the vaccine.
h. Table 6 can also be eliminated and only present the mean or median IgG concentration, according to the distribution of the variable, between the group without and with side effects from the vaccine.
i. Table 7, if the interest is only the group of events supposedly attributable to the first dose of the vaccine, this Table of results is not necessary.
j. Figures 2 and 3 can be eliminated, it is sufficient to report them in the results.
6. Discussion.
a. First paragraph please describe the main message of the study. Since the results of this study may be subject to change, the Discussion may also be changed, for the moment there are no further comments on this point.
Reviewer 2 Report
This is an interesting work focused on the presence if increased side-effects and higher antibody titers in those health care workers with previous COVID-19 infection who receive a first dose of a mRNA vaccine. The paper is of interested for clinicians, and for the journal. However, before considering its publications, I would recommend to address some minor comments.
The abstract is well-written. It summarizes the main aim of the paper, and collects all the information relevant in the results section. I recommend to add a few lines explaining which statistical analyses did the authors perform to analyse data on the first and the second dose of the mRNA vaccine.
The introduction section is really brief. How many types of vaccines are available? What are the differences between them in terms of side effects and titers of specific antibodies?
More details about statistical analysis would be helpful. Did the authors carry out logistic or linear regression models?
The authors found that individuals with a previous history of COVID-19 infection developed more vaccine-related side-effects and a higher anti-COVID-19 IgG titers compared to those individuals after who did not have history of COVID-19 infection. The main conclusion was that the first presented a more intense immune response. Could the authors extend this explanation by adding some hypotheses about humoral or cellular immune response? Why is this more intense immune response expected?
Reviewer 3 Report
The authors conduct an observational study and aimed to evaluate whether prior SARS-CoV-2 infection affects side effects and specific antibody production after vaccination with BNT162b2; this study included 1106 health care workers vaccinated with BNT162b2 at Tor Vergata University Hospital (Rome, Italy).
Comments:
The study is performed properly and the appropriate statistical analysis were used.
Round 2
Reviewer 1 Report
Dear Authors,
I have carefully read each of the responses to the comments made in relation to your manuscript and I am very grateful for the kindness in having considered some of them with the aim of improving the quality of the manuscript. However, there are some points on which my opinion will certainly differ from yours.
1. By the time the study was conducted, the hypothesis stating that the history of COVID-19 is associated with higher incidence of events supposedly attributable to the vaccine and higher immunogenicity to the BNT162b2 vaccine, has already been previously answered by several studies conducted around the world.
2. Given that the Title of the study states that only adverse reactions to the first dose of the BNT162b2 vaccine will be analyzed, I suggest that the entire discourse of the study be directed to this point.
3. In my opinion, the events supposedly attributable to the BNT162b2 vaccine should have been categorized as early and late and we should have been able to visualize whether or not this was related to the history of COVID-19.
4. It is clear that you wish to keep more Tables and Figures, in order to synthesize the content of the text; however, this causes the results to be repeated.
5. An important weakness has to do with the lack, not only of having identified whether or not the IgG antibodies have neutralizing activity, but also that the technique used to quantify them did not allow them to determine the real concentration of antibodies.
Round 3
Reviewer 1 Report
In my opinion the manuscript that was sent for review lacks novelty to date; there are already multiple studies that have delved into the frequency of early and late adverse events by the different anti-covid-19 vaccines. Likewise, reactogenicity has been evaluated several times.
The paper is likely to be accepted as a short communication, where the authors focus their attention on either adverse events or IgG antibody production.
Author Response
We thank the Reviewer for this comment.
Our article fulfills MDPI’s requirements for short communication.
We refer to the Editor for the final decision about the publication format.